# Predation risk of the sea urchin *Paracentrotus lividus* juveniles in an overfished area reveal system stability mechanisms and restocking challenges

**Federico Pinna**[1]*, **Nicola Fois**[2], **Francesco Mura**[2], **Alberto Ruiu**[3], **Giulia Ceccherelli**[1]

**1** Department of Chemical, Physical, Mathematical and Natural Sciences, University of Sassari, Sassari, Italy, **2** AGRIS Sardegna–Research Service for Fishery Products, Olmedo (SS), Italia, **3** Capo Caccia–Isola Piana Marine Protected Area, Alghero, (SS), Italia

* federicopinna27@gmail.com

## Abstract

Where sea urchin harvest has been so intense that populations have drastically regressed, concerns have arisen about the effectiveness of harvesting management. According to the theory of phase transition in shallow rocky reefs between vegetated and barren habitats, sea urchin recruitment, a key population structuring process, seems hampered by some stabilizing feedback despite an end to local human harvest of sea urchins. To shed a light on predation effects on sea urchin recruits, a 27-day field experiment was conducted using mega-predator exclusion cages (40x40x40 cm, 1 cm in mesh size) in barren and turf substrates. To facilitate this, 672 recruits (1.1 ± 0.02 cm in size) reared under control conditions were positioned in groups of 42 in each experimental unit (n = 4). Exclusion of mega-predators had a significant effect regardless the substrate, since a higher number of recruits was found under cages both in turf and barren. However, the results showed that in uncaged treatments the size of recruits that survived was larger in turf than in barren, as in the former substrate predation had reduced the abundance of the smallest recruits, highlighting that mega-predator presence affects differently the size of the recruits that had survived depending on the substrate. Overall, these results provide valuable information to address restocking actions of sea urchin populations in overharvested areas, where algal turfs are widespread, and assist studies on habitat stability mechanisms.

## 1. Introduction

Ecosystems, both marine and terrestrial, are characterized by complex interactions among species that can lead to significant species demographic fluctuations over time [1,2]. These changes, influenced by factors such as predation, diseases, and environmental conditions, have repercussions on community structure and functions [3,4]. Understanding these dynamics is crucial to address ecosystem management: if such variations concern key stone species that are

GitHub (Link: https://github.com/fpinna22/
Urchins_Survival).

**Funding:** The study has been funded by Ministry of
Agriculture Food and Forestry Policies (MIPAAF).
This study has also received funding from the
European Union Next-GenerationEU (PIANO
NAZIONALE DI RIPRESA E RESILIENZA (PNRR) –
MISSIONE 4 COMPONENTE 2, "Dalla ricerca
all'impresa"INVESTIMENTO 1.4 – D.D. 1034 17/06/
2022, CN00000033) and from Biodiversa+,
J13C21000150001. This manuscript reflects only
the authors' views and opinions, neither the
European Union nor the European Commission can
be considered responsible for them. The funders
had no role in study design, data collection and
analysis, decision to publish, or preparation of the
manuscript

**Competing interests:** The authors have declared
that no competing interests exist.The research was
conducted in the absence of any commercial or
financial relationships that could be construed as a
potential conflict of interest.

also resources, such understanding can fuel trade-offs between long-term conservation of the environment, sustainable exploitation, and economic viability.

An emblematic case of such dynamics is the regime shift observed in temperate rocky reefs caused by the grazing activity of sea urchins: these species can transform rich and diversified ecosystems into sterile barrens, having significant impacts on many other members of the benthic community *e.g.* [5–7]. In the Mediterranean infralittoral areas, the sea urchin *Paracentrotus lividus* (Lamarck 1816) inhabits both rocky reefs and *Posidonia oceanica* seagrass meadows [8], where it is considered one of the main herbivores and a keystone species for the indirect influence on the structure of the benthic community [9,10]. In fact, at high densities, *P. lividus* can transform heterogeneous algal communities into barren habitats, typified by bare rock and/or encrusting coralline algae [11–13]. This species plays a pivotal role in the trophic cascade that encompasses predatory fish, sea urchins, and macroalgae [14,15] and thus part of the propagation of top-down effects [5,6,16,17].

The primary predators of *P. lividus* are predominantly sparids (i.e., *Diplodus sargus*, *Diplodus vulgaris*, and *Sparus aurata*) and labrids (i.e., *Coris julis*, *Labrus merula*, *Thalassoma pavo*): they mainly feed on medium and small-sized sea urchins (around 4 mm), help to regulate the sea urchin population, sustaining the structure and functioning of coastal ecosystems [18–20]. Other predators capable of exerting control over *P. lividus* populations include small invertebrates such as crabs [20,21], gastropods (e.g., *Hexaplex trunculus*) [22] and sea stars (e.g., *Marthasterias glacialis*) [23]. These benthic predators are believed to exert negative feedback mechanisms on sea urchins, as in fact their very scarce abundance in barren areas would favor the settlement of urchin settlers, thereby increasing the recruitment and survival of the juveniles. Accordingly, the recovery of the vegetated state would thus be impeded despite the removal of adult individuals, maintaining the barren state [24].

Natural predation is not the only pressure exerted on *P. lividus*: as an edible species, human overharvesting has led to a marked decline in *P. lividus* populations across many European countries due to an increased demand stemming from the consumption of its gonads [25–27]. Therefore, *P. lividus* is now regarded as one of the most exploited benthic invertebrate species for commercial and recreational purposes [28,29]. In Sardinia (Italy) sea urchin harvest has been so intense that populations have drastically regressed, as highlighted by annual monitoring of *P. lividus* populations [30]: despite various restrictions imposed by the Sardinia Region to minimize the risk of over-exploitation [30], sea urchin populations have not recovered and concerns have arisen regarding the efficacy of harvest restrictions over the last several years [26]. Particular attention was dedicated to effectiveness of recruitment as if it was hindered by some stabilizing feedback mechanisms, and it could explain the lack of population recovery although sea urchin harvest has been managed and occasionally interrupted.

Generally, the population structure of sea urchins indeed depends on several factors acting during the settlement and post-settlement, such as larval dispersal, predation, and substrate heterogeneity [31–33]. The survival of early benthic invertebrate settlers, particularly in overexploited areas, often pose a bottleneck for their populations [34,35] and predation is likely the most significant cause of mortality even for juvenile sea urchins [23,36]. Predation, indeed, is a key selective force that acts upon the morphological and behavioral traits of the prey [37]. Therefore, the risk of predation can strongly influence the survival dynamics of *P. lividus* juveniles.

In this context, the primary concern is that the low density of adult sea urchins might hinder both the production of a substantial number of larvae and the reduction of turf and erect algal coverage (where small invertebrate predators are concentrated) [21] due to lack of an efficient grazing. Essentially, in the absence of adult individuals, the mechanisms facilitating the survival of the new generations would be lost, through various adaptive strategies such as physical protection [20,38,39], alteration of the algal community [9], and favorable settlement

signals (e.g. chemotaxis) [40]. This scenario would suggest that in over exploited areas the density of the adult sea urchins has gone over a critical threshold and the return to the population initial state might be hampered by a hysteresis effect [5,17].

To date, few studies have addressed the issue of sea urchin repopulation [41,42], and knowledge about restocking mechanisms remains largely unexplored. Therefore, whether the release into the wild of aquarium-reared young sea urchins could help to restore populations has to be explored and the conditions that improve the effectiveness need to be defined. This study consists of a manipulative experiment aimed to assess whether the survival of juvenile sea urchin individuals is influenced by the presence of mega-predators in barren and turf substrates. Specifically, it was questioned whether the predation risk of reared juvenile sea urchins may change depending on the substrate, and whether repopulation of *P. lividus* in macroalgae-dominated communities, such as turfs extensively present in shallow rocky reefs in Sardinia, is feasible. It was hypothesized that juvenile survival would be higher where mega-predators were excluded and in barren (rather than in turfs), where the abundance of adult sea urchins was higher and macro-predator abundance was lower. Results can provide crucial information to understand sea urchin ecology processes useful both to manage areas of exploitation and to address eventual repopulation efforts.

## 2. Materials and methods

### 2.1 *P. lividus* rearing under control conditions

In March 2022 adult specimens (n = 30, test diameter > 45 mm) were collected from a rocky substrate in a location within the Gulf of Alghero, in northwestern Sardinia, and transported to the laboratory. As a compensatory measure for the unavoidable collection of specimens, a reproductive strategy was implemented in the laboratory during the peak gonadal maturation period. This involved the extraction and utilization of gametes from freshly weighed gonads for the Gonadosomatic Index (GSI) analysis. Furthermore, a subset of these eggs was retained in the laboratory to facilitate controlled studies on larval development, settlement, and growth. A Bogorov tray was used to estimate egg or larval abundance. One milliliter of seawater containing fertilized eggs or larvae, diluted 1:10, was poured into the counting tray and its content was closely observed and counted under a dissection microscope. Ten counts of different samples were carried out and an average value*ml-1 calculated. The juvenile sea urchins utilized in our experiment were derived from this cohort, ensuring a sustainable approach to specimen utilization in the research. The gonads of the female specimens (19 units) were placed in a 500ml crystallizing dish with seawater, to facilitate the release and separation of the eggs. The eggs were then filtered using an 85 μm mesh filter and transferred to 5L beakers for subsequent fertilization. The sperm of the male specimens (11 units) were diluted in 500 mL of seawater and a few drops of this mixture (2 mL) were sufficient for the fertilization of the eggs.

During the incubation phase, the fertilized eggs were counted and carefully placed in a holding 20 L of seawater with reduced aeration for a duration of 24 hours; the density at this stage was adjusted to 300 larvae * mL$^{-1}$. Following hatching, the active larvae were counted and distributed into 250 L conical tanks at a density of 4 larvae * mL$^{-1}$.

The seawater used in the adult dissection, fertilization, and incubation phases had been previously filtered to 1 μm and sterilized with a UV lamp. The water temperature was maintained at 18˚C throughout the entire controlled reproduction process.

The larvae were fed a mix of phytoplankton (i.e. *Isochrysis galbana* and *Nannochloropsis oculata*). Larval survival was monitored every 3 days, coinciding with the feeding and partial water exchange. When the larvae reached the competent stage, they were transferred to 600 L tanks where both natural and artificial structures (oyster shells and granite and limestone tiles)

had been installed. These structures had been previously left in the lagoon for about 30 days to be colonized by epibionts. They provided a suitable substrate for larval settlement and offered a varied and natural diet to the post-metamorphic larvae. Once the larvae reached an exotrophic stage, they were fed with *Ulva lactuca* sourced from the wild. The juvenile individuals were kept in the settlement tanks in the laboratory until they reached a size of 15 mm, after which they were transferred to outdoor tanks until they reached the size suitable for the release into the wild (20 mm) [41].

### 2.2 Field experiment set up

The experimental site is located at Puntanegra (northwest of Alghero in Sardinia, Fig 1), which is an easily accessible site, where at 4 m of depth the rocky platforms are completely covered by erect fleshy and turf algae dominated by Dictyotales (see S1 Table, hereafter 'turf'). In fact, the abundance of the sea urchins *P. lividus* used to be quite high ($6.50\pm1.10$ m$^{-2}$, mean±SE n = 10), but because of the extensive and continuous harvest of the last 30 years at present it is extremely low ($0.86\pm0.16$ m$^{-2}$, n = 15).

To create areas of barren at the experimental site, on March 23, 2023, a total of 600 adult sea urchins were collected from Le Croci (south of Alghero), an unexploited site where adult *P. lividus* density was $8\pm1.4$ m$^{-2}$ and barren cover was $7.8 \pm 1.5\%$ (mean±SE n = 15). Sea urchins were transported (oxygenated tanks for a 4 hours) and released at the experimental site in an area of about 20.000 m$^{-2}$ at a density of 736g m$^{-2}$ (roughly 20 individuals m$^{-2}$) the estimated value that in oligotrophic regions allows the transition from vegetated habitat to barren state [7]. In a couple of months, the grazing of the introduced sea urchins triggered to several barren patches surrounded by turf algae.

Then, on May 25 (T0), 16 experimental units (approximately 1m$^2$ each) were chosen at a depth of about 4m: eight of them were on a barren substrate and eight were dominated by turf algae. For each substrate type (barren and turf) in four of the eight units a 40x40x40cm (1 cm of mesh size) exclusion cage for mega-predators was placed and in the remaining four, no cage

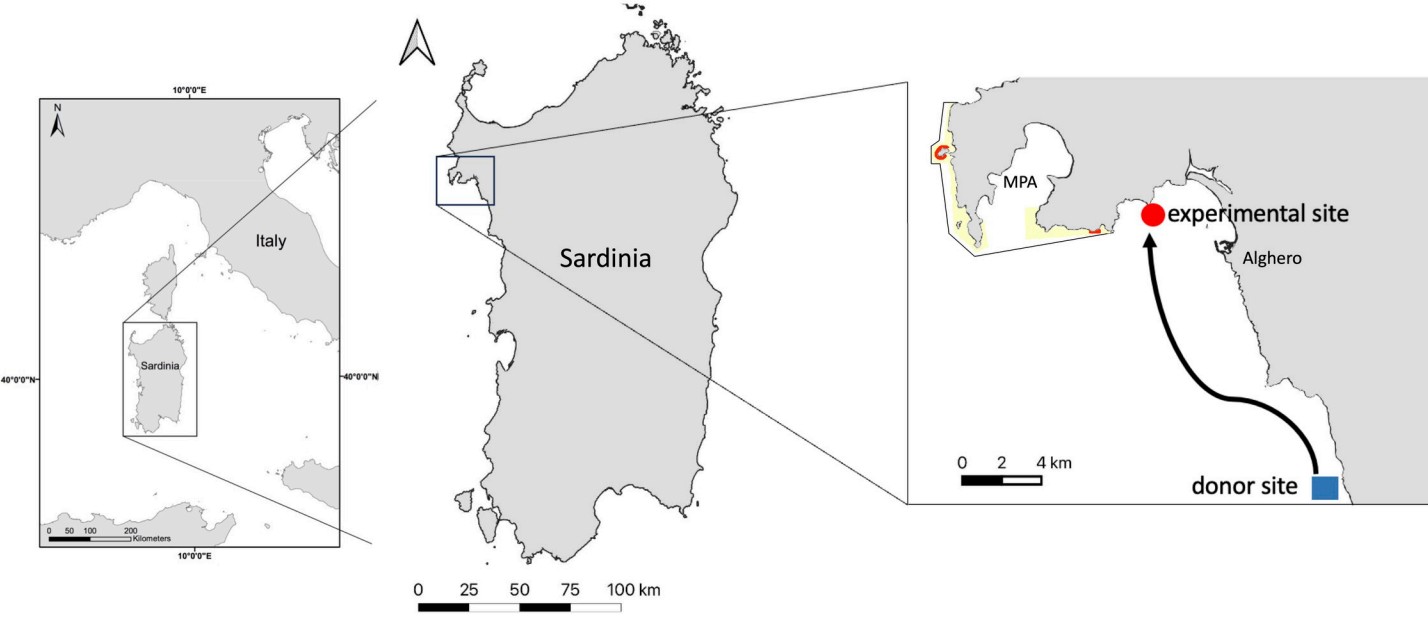

**Fig 1. Map of the study site.** Experimental site (Puntanegra) in red the sea urchin donor site (Le Croci) in blue.

was placed. Then, in each unit, 42 juvenile reared sea urchins were placed (a total of 672 individuals). The experimental design consisted of two factors, fixed and crossed: Substrate, with two levels (Turf and Barren), and Mega-Predators, with two levels (Yes and No, Fig 2).

Changes in the abundance of the juvenile sea urchins in the four treatment combinations were assessed for 27 days until the 20 of June (T1) by obtaining field count of the juvenile sea urchins with daily visits during the first week and then with a weekly frequency. Sampling was

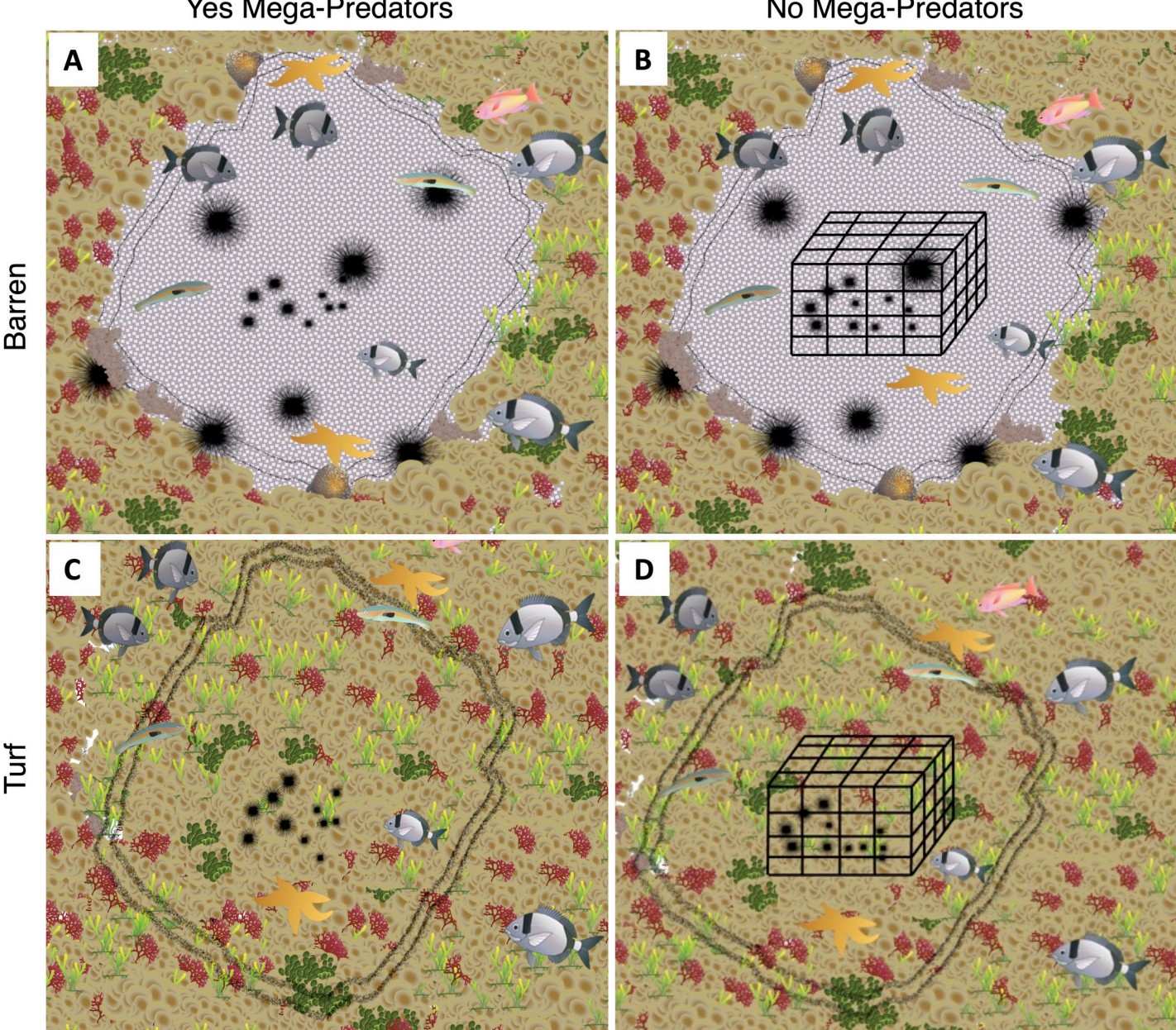

**Fig 2. Treatments used for the experiment.** (A) Barren substrate Mega-Predators yes; (B) Barren substrate Mega-Predators no; (C) Turf substrate Mega-Predators yes; (D) Turf substrate Mega-Predators no. Vector images used for the experimental design was adapted from the media library of the Center for Environmental Science (https://ian.umces.edu/media-library/symbols), under a CC BY license, with permission from University of Maryland Center of Environmental Science- Integration and Application Network, original copyright 2024.

meticulously conducted both within the designated experimental plot and in the immediate vicinity. This approach was adopted to identify the individuals that had moved outside the experimental area, although no individuals have ever found. Then, the predation risk was estimated by comparing the survival (%) of the juvenile sea urchins depending on the substrate (barren and turf) and mega-predator presence (Yes and No). The final survival rate at each experimental unit was described as the ratio between the number of days of survival and the total number of observation days, and the predation rate (proxy of the predation risk) was expressed as 1-survival rate (expressed on a scale ranging from 0 to 1).

To assess if the size of the juvenile sea urchins affected their probability of survival, on the initial and final days of the experiment all the individuals of each experimental unit were photographed and their individual size measured using image J.

## 2.3 Data analysis

The effect of the treatments on the abundance of the survived sea urchins at T1 was evaluated using a two-way univariate PERMANOVA [43] based on Euclidean distance, where Substrate (Barren and Turf), and Mega-Predators (Yes = no cages and No = cages) were fixed orthogonal factors. A posteriori, comparisons of means were done using Pair-Wise tests [43]. To characterize the differences in survival, the Kaplan-Meier product limit method [44] and the log-rank test [45] were used. These methods are commonly adopted in survival assays across diverse scientific disciplines as means to construct and statistically compare time-to-event data curves [22,46]. After gathering the time-to-event data, survival among substrate types and mega-predator presence was compared. Comparisons among the four treatment combinations were performed using Pair-Wise tests. The Cox proportional-hazards regression model (Coxph-test) was used to identify alternative hypotheses [47]: the model considered both substrate type and mega-predator presence. Findings were employed to ascertain predation risk and the contribution of each independent variable.

Additionally, the influence of the treatments on the size of the juvenile sea urchins was evaluated using a three-way univariate PERMANOVA [43] based on Euclidean distance, where time (T0 = initial and T1 = final), Substrate (Barren and Turf), and Mega-Predators (Yes = no cages and No = cages) were fixed orthogonal factors. A posteriori, comparisons of means were done using Pair-Wise tests [43].

PERMANOVAs and Pair-Wise tests were performed using PRIMER v7 software. Survival analyses and graphics were performed using Survival Analysis package [48] for R software using R version 4.2.2 [49].

## 2.4 Predator abundance

To estimate mega-predator abundance at the experimental site, underwater visual census (UVC) samplings were conducted on two sites hundreds of meters apart. Fish were sampled at approximately 5 m of depth along 25-m long and 5-m wide transects (n = 6) according to the 'strip transect' method [50]. Fish abundance was visually estimated by counting all specimens along the transect and attributing them to size classes of 2 cm.

A total of 21 species of fish were observed in the area surrounding the experimental site and, among these, all the sparids and labrids described as typical sea urchin predators were found (see S2 Table). Depending on the size (body length), the individuals were attributed to macro or mega predators: fish larger than 2 cm in length correspond to the mega-predator category, which was excluded by cages, except for *Coris juli*s, as its slender body allowed it to pass through the mesh of the exclusion cage even when the length exceeded 5 cm (FP personal observation) and they were treated as macro-predators. Regarding the mega-predators, in

addition to the fish, the presence of the sea star *Echinaster sepositus* (Retzius, 1783) needs to be acknowledged.

Furthermore, macrofauna predators (invertebrates) were sampled using a dredge on the 1 June, when two samples (each 400 cm$^2$) from each substrate type were taken, fixed in alcohol (75%) and sorted. A total of 29 and 12 macrofauna taxa were found in the turf and barren substrates, respectively: the taxonomic groups were differently represented among substrates, but potential predators were present in both substrates (See S3 Table).

## 3. Ethics statement

Although, the exploitation of the sea urchin *Paracentrotus lividus* is managed by Sardinian Region, approval from a research ethics committee was not required for the execution of this study, as the experiment was conducted in compliance with current regulation. Furthermore, the fieldwork did not involve any protected areas or areas subject to specific restrictions. The research was conducted entirely in public, non-protected marine environments where no specific permits for the collection of *P. lividus* or conducting research activities were necessary. This approach was in full conformity with local and national environmental regulations, ensuring that the study adhered to ethical principles of wildlife and habitat conservation.

## 4. Results

The abundance of the juvenile sea urchins has rapidly decreased through time at all four treatments and only few individuals were found at the end of the experiment. Unequivocally, the final abundance of the juvenile sea urchins was significantly higher where mega-predators were excluded respect to where they were present, regardless the substrate (Tables 1 and 3A). Particularly, the survival rate of the juvenile *P. lividus* individuals was overall low and largely affected by caging: since the sea urchin placement, the survival probability was less than 20% where mega-predators were excluded and below 5% in treatments with no exclusion cages (Fig 3B). However, significant differences in survival patterns were also found depending on the substrate (Fig 3B): indeed the Coxph test revealed that the interaction between the turf substrate and the presence of mega-predators is critical to the survival of the juvenile sea urchins (Table 2). Particularly, beyond the considerable impact of mega-predators on the juveniles, the substrate affected the predation rate of the sea urchin: in the caged treatments survival was significantly lower in the barren compared to the algal turf, likely due to the effect of less coverage by macroalgae (Fig 3B and Table 2).

However, also sea urchin size was affected by the interactive effect of treatments and in the turf substrate it changed through time where mega-predators were allowed (Table 3), since by the end of the experiment only the larger individuals (around 2 cm in size) survived (Fig 4). Conversely, in barren the presence of mega-predators did not influence the size of the survived juveniles, as in both caged and uncaged treatments the distribution of sea urchin size of the individuals was much more heterogenous (Fig 4 and Table 3).

**Table 1. Results of PERMANOVA on the abundance of sea urchin survived at T1 depending on the substrate (barren and turf) and mega-predator presence (yes and no).**

| Source | df | MS | Pseudo-F | P(perm) | P(MC) |
|---|---|---|---|---|---|
| Substrate (S) | 1 | 0.562 | 0.120 | 0.6852 | 0.7279 |
| Mega-Predators (MP) | 1 | 315.060 | 67.213 | **0.0001** | **0.0002** |
| SxMP | 1 | 7.562 | 1.613 | 0.2556 | 0.2263 |
| Res | 12 | 4.687 | | | |

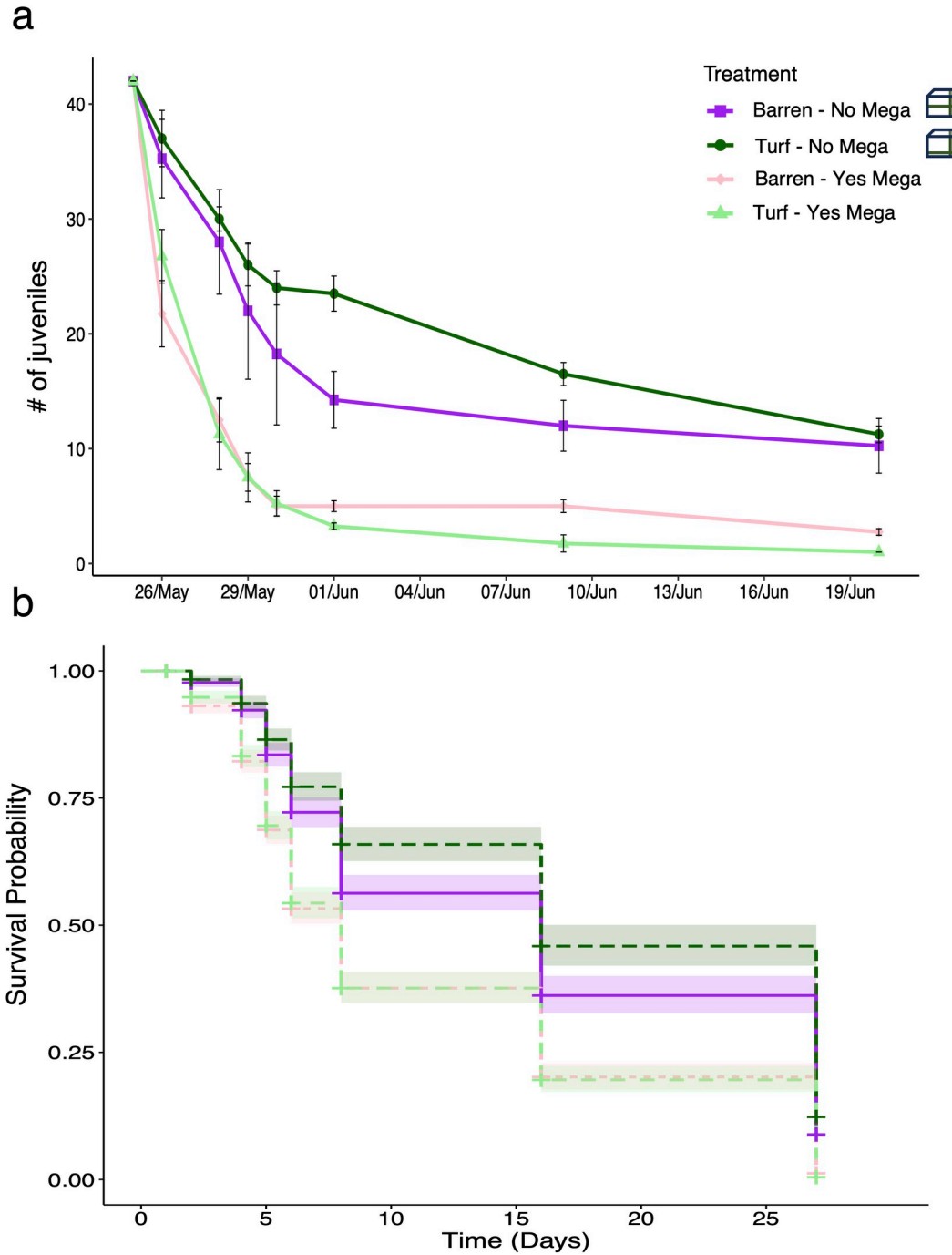

**Fig 3.** Abundance (upper) and survival (lower) of the *P. lividus* juveniles over the experimental time at all treatments. Significant differences were observed for *P. lividus* juveniles survival among treatments. Levels of significance are represented in Tables 1 and 2.

## 5. Discussion

The results gathered from this study highlight tangible challenges of eventual sea urchin restocking efforts since the survival probabilities observed for the reared juvenile sea urchins after the release into the wild were overall extremely low. However, exclusion of mega-

**Table 2. Predation risk of the juvenile sea urchins through time.** Significant Coxph-test results (above) between Substrate (barren and turf) and Mega-Predator presence (Yes and No). Log-Rank test results (below) comparing survival curves between treatments.

| | Coxph-test | | | | |
| --- | --- | --- | --- | --- | --- |
| | coef | exp(coef) | se(coef) | z | Pr(>\|z\|) |
| Turf | -0.22026 | 0.802 | 0.06019 | -3.660 | **0.000253** |
| Mega-Predators Yes | 0.53436 | 170.635 | 0.05193 | 10.289 | **< 2e-16** |
| Turf-Mega-Predators Yes | 0.23326 | 126.270 | 0.07569 | 3.082 | **0.002059** |
| | | | | | |
| **Kaplan Meier-Log-Rank Test Post hoc** | | | | | |
| | | Barren No | Barren Yes | Turf No | |
| Barren Yes | | **** | | | |
| Turf No | | *** | **** | | |
| Turf Yes | | **** | **n.s.** | **** | |

predators largely reduced the predation risk, as survival rate of *P. lividus* juveniles was significantly higher in caged units. This finding underscores the predominant role of large predators (both fishes and benthic species), at least on the short term, in regulating *P. lividus* recruitment. This result is particularly striking because the experimental site is located in an accessible coastal area where human activities are unrestricted where the abundance of fish mega-predators is generally lower and individuals are smaller than in Marine Protected Areas, depending on enforcement [51], as evidenced for the fish assemblage structure at Puntanegra compared to the nearby Capo Caccia Isola Piana MPA sites [52]. However, it is crucial to emphasize that the duration of our experiment (27 days) effectively narrows the range of possible mortality causes, making predation the predominant factor. Furthermore, during the study, we did not encounter skeletons or remains of deceased juvenile sea urchins; instead, we observed their absence in the experimental areas. This led us to the assumption that the absence of individuals was primarily due to predation. While we acknowledge that other mortality factors may exist, the nature and conditions of our experiment suggest that predation is the most likely cause of mortality in this specific context.

Nevertheless, the variation in sea urchin survival, albeit slight, between caged barren and caged turf treatments has provided insights about the little effect of the macrofauna predators on sea urchin juveniles, in the absence of megafauna. Even if the macro-predator assemblages

**Table 3. Results of PERMANOVA and significant pair-wise tests on the size of sea urchin depending on time (T0 and T1), substrate (barren and turf) and mega-predator presence (yes and no).**

| Source | df | MS | Pseudo-F | P(perm) |
| --- | --- | --- | --- | --- |
| Time = (T) | 1 | 3.614 | 12.580 | 0.0006 |
| Substrate = (S) | 1 | 3.007 | 10.465 | 0.0018 |
| Mega-Predators = (MP) | 1 | 0.938 | 3.267 | 0.0667 |
| TxS | 1 | 4.437 | 15.443 | 0.0002 |
| TxMP | 1 | 1.977 | 6.883 | 0.0098 |
| SxMP | 1 | 1.702 | 5.925 | 0.0141 |
| TxSxMP | 1 | 1.929 | 6.716 | **0.0107** |
| Res | 765 | 0.287 | | |
| Pair-Wise test on the interaction Time x Substrate x Mega-Predators | | | | |
| T1 Yes: Barren < Turf | | | | |
| T1 Turf: No < Yes | | | | |
| Turf Yes: T0 < T1 | | | | |

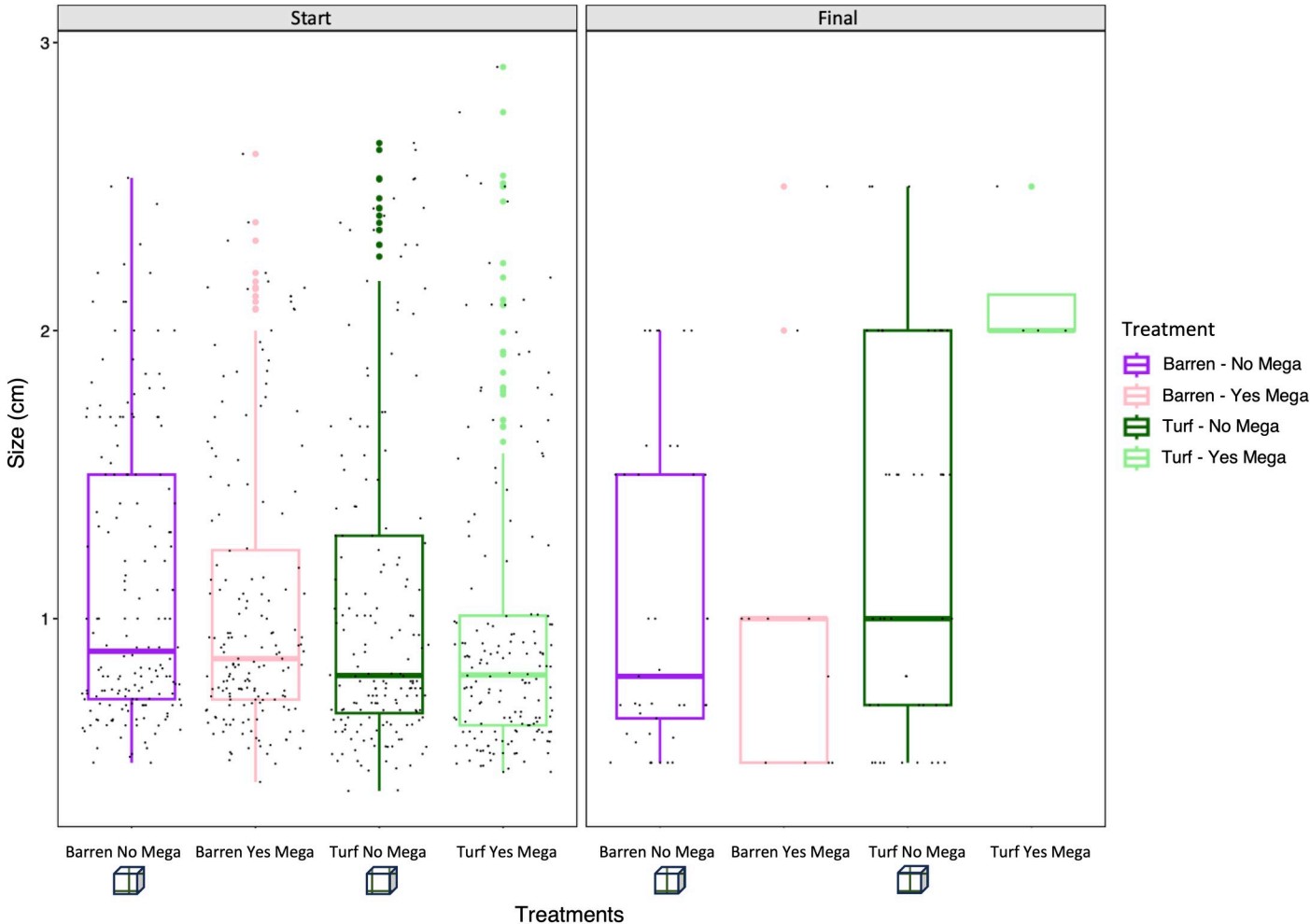

**Fig 4. Size distribution of *P. lividus* juveniles at the initial and final stage of the experiment.** Levels of significance are represented in Table 3.

are largely different between substrates, patterns in juvenile survival were very similar and size of the survived individuals was also unaffected, suggesting that macrofauna (small *Coris julis* included), certainly higher in turfs both in species and individual abundance, was not particularly efficient and size selective.

However, an evident interactive effect between substrate and mega-predators was found in sea urchin size, as in uncaged units the smallest sea urchin juveniles were completely depleted but only in turfs and not in barren. This result evidenced that the presence of mega-predators affected the juvenile sea urchins depending on their size and that in turfs the smaller their size, the higher the probability of being predated where megafauna is active. In barren, the same sized individuals (the smallest), have higher probability of survival. Particularly, in uncaged treatments, even though sea urchin survival rate was low, the protection provided by adults appeared to have a relevant influence on the smallest sized juveniles, since during sampling some of the juveniles were frequently found under the adults: the protective effect of the adult on the smallest sea urchin individuals that would reduce the predation risk was already thought [20,53,54] and would represent one of the proposed positive feed-back mechanisms to explain the higher sea urchin recruitment in barren [55]. The shelter effect provided by the adults might help interpreting why in the turf, where the adult sea urchins were lacking, only

the larger individuals survived in uncaged treatments. Therefore, the expectation of higher survival rate in the barren compared to the turf [5,21,56], was only partially met since survival rate was slightly lower through time, although on the short-term a similar abundance of survived juveniles was produced. However, observation of sea urchin individual behavior in the experimental units during sampling leads formulating hypotheses on different survival mechanisms that could potentially influence patterns of predation risk on the long-term. On this basis, the presence of adult sea urchin seems crucial to promote at least a direct positive influence on the survival of the juvenile sea urchins.

Consistently, the prediction of a lower probability of sea urchin survival in turfs due to a higher predation risk was here met for the smallest juveniles. This phenomenon represents the basis to explaining the stability of the vegetated substrate and that involves cascading effects of the whole trophic chain: specifically, experimental evidence has been provided that algal system stability relies on the high abundance of urchin predators of a wide size range that protect these habitats from sea urchin grazing [21,57,58]. In fact, macrophyte-formed habitats are usually characterized by high biodiversity and high production rates, likely proportional to their physical complexity [59–61]. Therefore, in rocky subtidal habitats, transitions between barrens to algal dominated systems have been described as consequence of either reestablished top-down control on sea urchins for a marine reserve effect [51], sea urchin overharvesting [62], storms and diseases [63,64]. However, most of the effort has been focused at understanding the mechanisms that stabilize algal forests [5,65,66], likely because of their high ecological role (*i.e.* foundation species), and much less attention has been devoted to lower complex algal systems such as algal turfs. These latter are increasingly reported to be globally expanding at the expense of kelps and canopy-forming algae and are used as indicators of human-made disturbances [67–69]. In fact, common coastal stressors such sedimentation and nutrient availability are reported to positively affect algal turfs [70–72]. Additionally, on reefs exposed to humans, overfishing and thus predator depletion is weakening the top-down control necessary for mitigating the increasing algal abundance [62,73].

Therefore, the evidence gained in this study, supported by the theory on the stability of algal dominated communities, indirectly suggesting that natural recruitment of *P. lividus* in the over-harvested Sardinia areas will be largely prevented. Some other concerns will also be raised about the effectiveness of *P. lividus* restocking through rearing of juvenile individuals whose mortality, once released into the wild, should be given for granted due to the presence of mega-predators even in unprotected areas, where human activities are unrestricted. Findings have also highlighted that adult sea urchin presence would enhance juvenile survival. Nevertheless, several questions remained unanswered, such as to what extent the predation risk depends on the adult sea urchin specific density [74] or if artificial structures providing sea urchin shelters could enhance juvenile survival, compensating the protective role of the adults, in assisted natural recovery efforts [75]. However, lessons learnt by the difficulties in *Diadema antillarum* sea urchin restoration at the Caribbean Sea after massive die-off since 1983 [75]. should warn on the objective difficulties in modifying the feedback mechanisms of stability, which are likely to be complex. Overall, results have highlighted that a much deeper knowledge about processes regulating sea urchin recruitment prevention would be essential before targeted approaches can be developed to promote the repopulation of depleted sea urchin populations.

## Supporting information

**S1 Table. List of algae composing turfs in experimental area.**
(DOCX)

**S2 Table. Average abundance (± SE) in 125 m² of fish in the experimental area depending on their size: Individuals smaller (macro-predators, that potentially enter the cages) and larger (mega-predators) than 2 cm in body length.** The asterisk indicates the predators of *Paracentrotus lividus* (Sala, 1997; Guidetti 2004; Bonaviri et al., 2010).
(DOCX)

**S3 Table. List of macrofauna species found in both Turf substrate and Barren substrate in the experimental area.** The asterisk indicates the general carnivores.
(DOCX)

## Acknowledgments

We are also grateful to Ivan Guala, Gabriele Costa, Arianna Pansini, Alessandra Puccini and Patrizia Stipich their field survey support and Luigi Piazzi and Marco Curini Galletti for their support in algal and macrofauna species identification, respectively.

## Author Contributions

**Conceptualization:** Federico Pinna, Nicola Fois, Giulia Ceccherelli.

**Data curation:** Federico Pinna.

**Formal analysis:** Federico Pinna.

**Funding acquisition:** Nicola Fois, Giulia Ceccherelli.

**Investigation:** Federico Pinna, Francesco Mura, Alberto Ruiu.

**Methodology:** Federico Pinna, Francesco Mura, Giulia Ceccherelli.

**Project administration:** Giulia Ceccherelli.

**Resources:** Nicola Fois, Giulia Ceccherelli.

**Software:** Federico Pinna.

**Supervision:** Giulia Ceccherelli.

**Validation:** Federico Pinna, Nicola Fois, Francesco Mura, Alberto Ruiu, Giulia Ceccherelli.

**Visualization:** Federico Pinna.

**Writing – original draft:** Federico Pinna.

**Writing – review & editing:** Federico Pinna, Giulia Ceccherelli.

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
