## [Decision Letter · Decision Letter 0]

22 Jan 2024

PONE-D-23-40934Predation risk of the sea urchin Paracentrotus lividus juveniles in an overfished area reveal system stability mechanisms and restocking challengesPLOS ONE

Dear Dr. Pinna,

Thank you for submitting your manuscript to PLOS ONE. After careful consideration, we feel that it has merit but does not fully meet PLOS ONE’s publication criteria as it currently stands. Therefore, we invite you to submit a revised version of the manuscript that addresses the points raised during the review process.

We look forward to receiving your revised manuscript.

Kind regards,

Tobias B. Grun, Ph.D.

Academic Editor

PLOS ONE

“The study has been funded by Ministry of Agriculture Food and Forestry Policies (MIPAAF).   This study has also received funding from the European Union Next-GenerationEU (PIANO NAZIONALE DI RIPRESA E RESILIENZA (PNRR) – MISSIONE 4 COMPONENTE 2, "Dalla ricerca all'impresa"INVESTIMENTO 1.4 – D.D. 1034 17/06/2022, CN00000033). This manuscript reflects only the authors’ views and opinions, neither the European Union nor the European Commission can be considered responsible for them.”

3. In the online submission form, you indicated that [Data will be made available on request.].

5. We note that Figure 2 in your submission contain copyrighted images. All PLOS content is published under the Creative Commons Attribution License (CC BY 4.0), which means that the manuscript, images, and Supporting Information files will be freely available online, and any third party is permitted to access, download, copy, distribute, and use these materials in any way, even commercially, with proper attribution. For more information, see our copyright guidelines: http://journals.plos.org/plosone/s/licenses-and-copyright.

1. You may seek permission from the original copyright holder of Figure 2 to publish the content specifically under the CC BY 4.0 license.

Reviewers' comments:

Reviewer's Responses to Questions

**Comments to the Author**

1. Is the manuscript technically sound, and do the data support the conclusions?

Reviewer #1: Yes

2. Has the statistical analysis been performed appropriately and rigorously? 

Reviewer #1: Yes

3. Have the authors made all data underlying the findings in their manuscript fully available?

Reviewer #1: No

4. Is the manuscript presented in an intelligible fashion and written in standard English?

Reviewer #1: Yes

5. Review Comments to the Author

Reviewer #1: The study presented seeks to explore the relationships between juvenile Paracentrodus lividus recruitment and various predation pressures within different habitats, with the goal of improving the management of this heavily exploited taxon. This experimental study used tank raised juvenile P. lividus to comparatively assess survival rates in heavily vegetated (turf) and non-vegetated (barrens) natural habitats, where megapredators (length > 2 cm) were either present or excluded (via 40x40x40 cm cages w/ 1 cm mesh). Overall, the work performed by the authors appears to report significant results that should be of interest to a broad readership. The methodology is sound, and the interpretations and conclusions are well supported by the experimental data. There are some small issues regarding clarity and grammatical errors (see minor suggestions below), but the undisclosed restrictions regarding data availability are of main concern. Assuming that those issues (especially data availability) are adequately addressed, I support publication of the revised version of this manuscript in PLoS ONE.

Major Comments:

1. Data Availability

a. Authors answered “No – some restrictions will apply” regarding data being fully available without restriction. It would pay well to elucidate the restrictions, and what this means regarding the ability to replicate the experiment and/or data analyses.

Minor comments: Clarity Suggestions

1. Introduction

a. Lines 72–73 – “Accordingly, the recovery of the vegetated state would thus be impeded despite the removal of adult individuals, maintaining the barren state” – is unclear what the authors are trying to convey, perhaps a line along the lines of – “The lower predation rates of juvenile urchins would therefore hinder the recovery of the vegetated state and maintain the barrens, even in the absence of adult urchins”.

2. Material

---

## [Author Response · Author response to Decision Letter 0]

26 Feb 2024

Dear Editor,

we are thankful for the time and effort you have devoted to processing our manuscript. We have appreciated the comments and indication of the Reviewer and we have now completed the revision of the manuscript. Firstly, below we provide our detailed responses to each of the points raised regarding the journal requirements.

We confirm that our manuscript has been formatted in accordance with the templates and style guidelines required by PLOS ONE. 

Additionally, we affirm that the funders had no role in study design, data collection and analysis, decision to publish, or preparation of the manuscript. 

Regarding data availability, we are pleased to inform you that all data underlying the findings described in our manuscript will be shared and made publicly accessible. The data will be available at the following link: https://github.com/fpinna22/Urchins_Survival in accordance with the open access policy of PLOS ONE.

Regarding Figure 1, which shows a map of the study area, we wish to clarify that it was generated using vector images created by QGIS software version 3.28. Consequently, there are no external copyright issues requiring authorization. 

Regarding Figure 2, we confirm that it was created using vector images sourced from the IAN/UMCES Symbol and Image Libraries (https://ian.umces.edu/media-library/symbols/#attribution), which allow the free use of such resources with proper attribution, as specified. These images are licensed under the Creative Commons Attribution-ShareAlike 4.0 International (CC BY-SA 4.0), and the required attribution has been included in the caption of figure 2 (Lines 215- 218).

We have revised the text accordingly to the comments received. Down below you you will find the point-by-point responses. We hope that the responses provided are satisfactory and that the revisions made to the manuscript meet your requests. We sincerely hope this version will be deemed worthy of publication in PLOS ONE.

Best regards

Federico Pinna

The study presented seeks to explore the relationships between juvenile Paracentrotus lividus recruitment and various predation pressures within different habitats, with the goal of improving the management of this heavily exploited taxon. This experimental study used tank raised juvenile P. lividus to comparatively assess survival rates in heavily vegetated (turf) and non-vegetated (barrens) natural habitats, where megapredators (length > 2 cm) were either present or excluded (via 40x40x40 cm cages w/ 1 cm mesh). Overall, the work performed by the authors appears to report significant results that should be of interest to a broad readership. The methodology is sound, and the interpretations and conclusions are well supported by the experimental data. There are some small issues regarding clarity and grammatical errors (see minor suggestions below), but the undisclosed restrictions regarding data availability are of main concern. Assuming that those issues (especially data availability) are adequately addressed, I support publication of the revised version of this manuscript in PLoS ONE.

Major Comments:

1. Data Availability

a. Authors answered “No – some restrictions will apply” regarding data being fully available without restriction. It would pay well to elucidate the restrictions, and what this means regarding the ability to replicate the experiment and/or data analyses.

RESP: We agree. The raw data supporting the conclusions of this article will be available in the public repository GitHub at the following link: https://github.com/fpinna22/Urchins_Survival

Minor comments: Clarity Suggestions

1. Introduction

a. Lines 72–73 – “Accordingly, the recovery of the vegetated state would thus be impeded despite the removal of adult individuals, maintaining the barren state” – is unclear what the authors are trying to convey, perhaps a line along the lines of – “The lower predation rates of juvenile urchins would therefore hinder the recovery of the vegetated state and maintain the barrens, even in the absence of adult urchins”. 

RESP: Actually, the scientific literature acknowledges that one of the feedback mechanisms stabilizing the barren state is the lower occurrence of macrofauna which exerts predatory pressure on P. lividus juveniles. Consequently, a lower abundance of macrofauna correlates with a higher sea urchin recruitment and survival in barren substrates that lead to increased grazing pressure, which in turn reduces macroalgal communities and stabilizes barrens state. We have not made changes in the text.

2. Materials and Methods

a. Lines 135 – 137 – How were fertilized eggs and larvae counted during the incubation and raising of the juvenile urchins? Elucidating should only improve method clarity.

RESP: Done. Thank you for highlighting the need for clarity in our method description. Specifically, we added the following details. Lines 138- 141 “A Bogorov tray was used to estimate egg or larval abundance. One milliliter of seawater containing fertilized eggs or larvae, diluted 1:10, was poured into the counting tray and its content was closely observed and counted under a dissection microscope. Ten counts of different samples were carried out and an average value*ml-1 calculated”.

Lines 151 – 152 – “they were transferred to outdoor tanks until they reached the size suitable for the release into the wild” – it is unclear as to what the designated “suitable” size is, a 15 mm size is mentioned for transition from indoor to outdoor tanks, but nothing about outdoor tanks to the field. Correction of this would improve ability to replicate the study. 

RESP: Done. We agree with the comment. We added the suitable size and cited a reference that clarify the method utilized. It specifies that juvenile urchins were transferred to outdoor tanks and monitored until they reached a minimum size of 20 mm, which, based on Giglioli et al. (2021), is considered optimal for their survival upon release. Line 166.

b. Lines 177 – 178 – “by counting in the field the juvenile sea urchin with daily visits during the first week and then with a weekly frequency.” – How were juvenile sea urchins in sites without predator exclusion cages kept within the study area? Correction of this would improve method clarity and the ability to replicate the study.

RESP: Done. Thank you for the comment. We added a sentence in the text to clarify our methodology: Line 178-182 “Sampling was meticulously conducted both within the designated experimental plot and in the immediate vicinity. This approach was adopted to identify the individuals that had moved outside the experimental area, although no individuals have ever found”.

c. Lines 182 – 183 – “and the predation rate (proxy of the predation risk) was expressed as 1-survival rate (expressed on a scale ranging from 0 to 1).” – how can we assume that predation is the only form of mortality in these juvenile urchins? – elucidation would improve method clarity.

RESP: Done. Thank you for highlighting this point. We have incorporated a sentence in the Discussion section (Line 313-320): “However, it is crucial to emphasize that the duration of our experiment (27 days) effectively narrows the range of possible mortality causes, making predation the predominant factor. Furthermore, during the study, we did not encounter skeletons or remains of deceased juvenile sea urchins; instead, we observed their absence in the experimental areas. This led us to the assumption that the absence of individuals was primarily due to predation. While we acknowledge that other mortality factors may exist, the nature and conditions of our experiment suggest that predation is the most likely cause of mortality in this specific context”. 

Minor Comments: Grammatical/Typological Suggestions

1. Abstract

a. Line 30 – “a key process structuring population” – suggested replacement – “a key population structuring process”.

RESP: Done

b. Line 31 – “despite sea urchin human harvest has quit” – suggested replacement – “despite an end to local human harvest of sea urchins”.

RESP: Done

c. Line 33 – “At this aim” – suggested replacement – “To facilitate this”.

RESP: Done

d. Line 37 – “survived recruits” – suggested replacement – “the recruits that survived”.

RESP: Done

e. Line 38 – “has” – suggested replacement – “had”.

RESP: Done

f. Line 40 – “survived recruits depending on the substrate” – suggested replacement – “the recruits that had survived, depending on the substrate”.

RESP: Done

2. Introduction

a. Line 76 – “countries,” – suggested replacement – “countries”.

RESP: Done

b. Line 82 – “last years” – suggested replacement – “over the last several years”.

RESP: Done

3. Materials and Methods

a. Line 177 – “by counting in the field the juvenile sea urchin” – suggested replacement – “by obtaining field counts of the juvenile sea urchins”.

RESP: Done

---

## [Editor Report · Decision Letter 1]

12 Mar 2024

Predation risk of the sea urchin Paracentrotus lividus juveniles in an overfished area reveal system stability mechanisms and restocking challenges

PONE-D-23-40934R1

Dear Dr. Pinna,

We’re pleased to inform you that your manuscript has been judged scientifically suitable for publication and will be formally accepted for publication once it meets all outstanding technical requirements.

Kind regards,

Tobias B. Grun, Ph.D.

Academic Editor

PLOS ONE

---

## [Editor Report · Acceptance letter]

8 Apr 2024

PONE-D-23-40934R1 

PLOS ONE

Dear Dr. Pinna, 

I'm pleased to inform you that your manuscript has been deemed suitable for publication in PLOS ONE. Congratulations! Your manuscript is now being handed over to our production team.

Kind regards, 

on behalf of

Dr. Tobias B. Grun 

Academic Editor

PLOS ONE